# Antimicrobial Susceptibility Testing for Salmonella Serovars Isolated from Food Samples: Five-Year Monitoring (2015–2019)

**DOI:** 10.3390/antibiotics9070365

**Published:** 2020-06-29

**Authors:** Maria Francesca Peruzy, Federico Capuano, Yolande Thérèse Rose Proroga, Daniela Cristiano, Maria Rosaria Carullo, Nicoletta Murru

**Affiliations:** 1Department of Veterinary Medicine and Animal Production, University of Naples “Federico II”, Via Delpino 1, 80137 Naples, Italy; mariafrancesca.peruzy@unina.it (M.F.P.); murru@unina.it (N.M.); 2Department of Food Microbiology, Istituto Zooprofilattico Sperimentale del Mezzogiorno, Via Salute, 2, 80055 Portici (Naples), Italy; federico.capuano@cert.izsmportici.it (F.C.); daniela.cristiano@izsmportici.it (D.C.); mariarosaria.carullo@cert.izsmportici.it (M.R.C.)

**Keywords:** antimicrobic resistance, *Salmonella* serovars, *S.* Infantis, foods

## Abstract

The continuous collection and analysis of updated data on the antimicrobic resistance among bacterial strains represent the essential core for the surveillance of this problem. The present work aimed to investigate the occurrence of antimicrobial resistance among *Salmonella* serovars isolated in foods in 2015–2019. A total of 178 *Salmonella* strains belonging to 39 serovars were tested against 10 antimicrobials. High proportions of *Salmonella* isolates were resistant to tetracycline (*n* = 53.9%), ciprofloxacin (*n* = 47.2%), ampicillin (*n* = 44.4%), nalidixic acid (*n* = 42.7%), and trimethoprim-sulfamethoxazole (*n* = 38.8%). Different resistance rates were recorded among the different serotypes of *Salmonella,* and *S.* Infantis, exhibited the highest resistance to antibiotics. A high percentage of strains isolated from poultry, pork, and bovine were resistant to at least one or two antimicrobials. Resistant and multidrug-resistant (MDR) strains were also recorded among the isolates from molluscan shellfish; however, the occurrence of resistant *Salmonella* strains isolated from this source was significantly lower compared with those reported for poultry, pork, and bovine. The high levels of resistance reported in the present study indicate a potential public health risk. Consequently, additional hygiene and antibiotic stewardship practices should be considered for the food industry to prevent the prevalence of *Salmonella* in foods.

## 1. Introduction

In 2018, salmonellosis, mainly due to *Salmonella enterica*, was the second most commonly reported bacterial foodborne zoonoses in the European Union, with 91,857 confirmed cases [1]. *S. enterica* is a Gram-negative bacterium with more than 2600 serotypes that, based on their different pathogenic behaviors, can be divided into two groups: typhoidal *Salmonella* and nontyphoidal *Salmonella* [2]. While typhoidal *Salmonella* is associated with a high number of fatal cases [3], nontyphoidal *Salmonella* infections in humans are generally self-limiting and do not require antimicrobial treatment [4]. However, in rare cases, the infection can be more serious, and the use of antimicrobial agents such as fluoroquinolones and third-generation cephalosporins, generally recommended for treating both adults and children, is essential. *Salmonella* as well as other pathogenic bacteria can exhibit resistance to a wide range of antibiotics, and it has been demonstrated that multidrug-resistant *Salmonella* infection may have a more serious human health impact compared to infection by less resistant strains [4]. According to the World Health Organization (WHO), antimicrobic resistance (AMR) is one of the most important public health threats of the 21st century. Globally, antibiotic-resistant bacteria (ARB) already cause more than 70,000 deaths each year, and it has been predicted that, in the near future, this problem will involve millions of people throughout the world [5]. Antibiotic resistance plays an important role in the increased incidence of different bacterial infections and the continuous collection and analysis of data on AMR—in particular, understanding the degree of antibiotic resistance in different bacterial species is essential for the planning, implementation, and evaluation of public health practices [6]. ARB may reach humans following their direct contact with infected animals or biological substances such as blood, saliva, milk, feces, or urine (direct exposure) or through their consumption of contaminated foods such as eggs, meat, and dairy products (indirect exposure) [7]. The presence of antibiotic-resistant bacteria in the food chain could be attributed to the use of antibiotics in aquaculture, livestock production, and crop culture or to the spread of resistant bacteria from the environment at any step of the food production chain. Moreover, ARB may also enter the marine environment and thus contaminate marine animals. For AMR in *Salmonella,* a strong association between resistant bacteria from human cases and those from food-producing animals was demonstrated [8]. In general, AMR levels for *Salmonella* isolated either from human cases or food samples are influenced by serovars. The five most commonly reported serovars in food are *S.* Enteritidis, *S.* Typhimurium, monophasic *S.* Typhimurium, *S.* Infantis, and *S*. Derby, while these along with *S.* Newport, *S.* Stanley, *S.* Kentucky, *S.* Virchow, and *S.* Agona are frequently reported in human cases [1]. The different *Salmonella* serovars can be found in a wide range of foods such as poultry, fish, eggs, beef, and dairy products [9]. However, while *S.* Typhimurium shows an ubiquitous distribution, other serovars such as *S.* Derby, mainly isolated from pork and pork products, are strictly associated with a food category [10]. Resistance markers can easily be transferred among bacteria belonging to the same or different species; thus, the monitoring of AMR in serovars less clinically important is also crucial to tracking early changes in the microbial population [2]. 

Antimicrobial resistance patterns may change rapidly over time, and prompt detection of these variations through the collection of updated data is essential to quickly change national and European treatment guidelines. Therefore, the present work aimed to investigate the occurrence of antimicrobial resistance among *Salmonella* serovars isolated from foods in 2015–2019. 

## 2. Results

The antibiotic susceptibility of tested strains is shown in Figure 1. High proportions of *Salmonella* isolates were resistant to tetracycline (*n* = 96.539%), ciprofloxacin (*n* = 84.472%), ampicillin (*n* = 79 444%), nalidixic acid (*n* = 76.427%), and trimethoprim-sulfamethoxazole (*n* = 69.388%), while moderate resistance was recorded toward cefotaxime (17.4%), chloramphenicol (10.1%), and ceftazidime (6.2%). To eliminate the influence of the high prevalence of *S.* Infantis (59 out of 178 isolates), such evaluation was also carried out after excluding the *S.* Infantis serotype (Figure 1). Comparing the overall resistance of all serotypes with those without *S.* Infantis, significant differences were observed in the prevalence of strains resistant to tetracycline, ciprofloxacin, nalidixic acid, trimethoprim-sulfamethoxazole, and cefotaxime. 

Seventy-five strains (42.1%) out of the 178 isolates were resistant to one or two antimicrobial classes, while 74 (41.6%) were multidrug resistant (MDR), and one strain was resistant to all antibiotics tested. Moreover, the 178 *Salmonella* strains analyzed showed 45 different patterns of resistance. In particular, 25 strains (14.04%) were simultaneously resistant to ampicillin, trimethoprim-sulfamethoxazole, ciprofloxacin, and nalidixic acid. Moreover, 65 strains (36.5%) showed coresistance to both the fluoroquinolones tested, and 25 strains (14%) showed coresistance to ciprofloxacin and cefotaxime. Only 29 strains (16.3%) showed susceptibility to all antibiotics considered.

The comparison between the sampling periods 2015–2016 and 2017–2019 showed an increase in resistance to four single drugs tested (tetracycline, ciprofloxacin, nalidixic acid, and trimethoprim-sulfamethoxazole) (Figure 2). In particular, the occurrence of resistance to ciprofloxacin between the two sampling periods showed a significant difference (*p* < 0.05). No differences were observed in the occurrence of MDR strains or the coresistance to ciprofloxacin and cefotaxime, while the number of isolates resistant to one or two antimicrobial classes was significant higher (*p* < 0.05) in the 2017–2019 period (2015–2016 = *n*. 11; 2017–2019 = *n*. 67).

By serovar, resistance to tetracycline (TET) and ampicillin (AMP) was extremely high in *S.* Newport (TET = 100%, AMP = 100%), *S.* Brandenburg (TET = 100%, AMP = 80%), *S.* Infantis (TET = 89.7%, AMP = 63.8%), monophasic *S.* Typhimurium (TET = 77.8%, AMP = 94.4%), and *S.* Rissen (TET = 75%, AMP = 75%) (Figure 3). 

*S.* Infantis also showed an extremely high level of resistance to nalidixic acid (98.3%) and ciprofloxacin (84.5%) and, along with *S.* Brandenburg and *S.* Rissen, to trimethoprim-sulfamethoxazole (*S.* Infantis = 88%, *S.* Brandenburg = 75%, and *S.* Rissen = 75%). Moreover, the occurrence of resistance to ciprofloxacin among the *S.* Infantis strains from 2015 to 2019 showed a significant increase (*p* < 0.05). Moreover, 49 (83.05%) among the *S.* Infantis strains were coresistant to the two fluoroquinolones tested. For the investigated serovars, MDR was most frequently reported among *S.* Newport (100%), followed by *S.* Brandenburg (75%), *S.* Rissen (75%), and *S.* Infantis (71.2%). Furthermore, one strain among the monophasic *S.* Typhimurium isolates was resistant to all the antimicrobials tested.

By source, microbiological resistance to the antimicrobials tested, excluding colistin sulfate and gentamicin, ranged from high to extremely high among *Salmonella* spp. from poultry (Table 1). 

The two closely related *Salmonella* serovars, *S.* Typhimurium and monophasic *S.* Typhimurium, showed resistance profiles substantially equal except for their resistance toward ampicillin and tetracycline. Indeed, monophasic *S.* Typhimurium showed and higher and statistically significant resistance toward both the antibiotics (AMP 94% vs. 36%, X2 = 9.8, *p* < 0.002; TET 78% vs. 18%, X2 = 11.5, *p* < 0.0007).

Considering *Salmonella* spp. data from bovine and pork, overall resistance to ampicillin, trimethoprim-sulfamethoxazole, tetracycline, and fluoroquinolones was high. Moderate levels of resistance to the compounds tested were detected among isolates from molluscan shellfish, except for ampicillin and tetracycline, for which the resistance was high (Table 1). Significant differences in the occurrence of resistance to the antimicrobials tested were observed between *Salmonella* spp. isolates from molluscan shellfish and the other sources (*p <* 0.05). Among isolates from buffalo, *S.* Muenchen was resistant to both fluoroquinolones tested, and *S.* Stanleyville was resistant to ampicillin and trimethoprim-sulfamethoxazole. Concerning isolates from vegetables, two strains (of serovars Nottingham and Hvittingfoss) showed susceptibility to all antibiotics considered, while *S.* Kasenyi was resistant to gentamicin and, along with *S.* Winston, to ciprofloxacin. One strain out of three isolated from snail was resistant to ampicillin. The levels of MDR and resistance to one or two antimicrobial classes among *Salmonella* isolates from different sources are shown in Figure 4. The occurrence of MDR strains isolated from poultry was significantly higher compared to MDR strains isolated from other sources (*p <* 0.05). 

## 3. Discussion

In the present study, the antimicrobial resistance of *Salmonella* serotypes isolated from different sources was tested against 10 antimicrobials, and 45 different patterns of resistance were recorded, confirming the wide diversity of resistance profiles in *Salmonella* spp. isolated from foods.

The highest levels of resistance were demonstrated against tetracycline, fluoroquinolones (ciprofloxacin and nalidixic acid), ampicillin, and trimethoprim-sulfamethoxazole. These high levels of resistance, consistent with other studies [4,11,12], are of particular concern since these drugs are used as first-line treatment for infection in humans or animals. In particular, fluoroquinolones are the gold standard for treatment against invasive salmonellosis in humans, and ampicillin, sulfamethoxazole, and tetracycline are widely used in veterinary medicine as first-line treatment in animal infections [13]. However, fluoroquinolones, categorized according to the World Health Organization (WHO) as “highest priority critically important antimicrobials” (HPCIA), can also be used in veterinary medicine when there are no alternative antimicrobials [13]. Moreover, 36.5% of isolates tested showed coresistance to both fluoroquinolones, and around 14% of the strains showed a simultaneous resistance to ciprofloxacin and nalidixic acid, ampicillin, and trimethoprim-sulfamethoxazole. These latter results are of particular concern because amoxicillin and trimethoprim-sulfamethoxazole are used as second-line therapies in humans who fail to respond to the first-line antibiotics (e.g., in case of infection caused by resistant bacteria) and in those with persistence of symptoms [14].

When fluoroquinolones are not recommended (e.g., during treatment of children infection), the “critically important antimicrobials” (CIA) third-generation cephalosporins are the antimicrobials of choice for the treatment of human *Salmonella* infections. The resistance to these compounds in *Salmonella* spp. analyzed was moderate (cefotaxime = 17.4% and ceftazidime = 6.2%); however, it was higher than that reported by other authors (cefotaxime = 6.5% and ceftazidime = 2.3% [12]; cefotaxime = 1.3% and ceftazidime = 0.0% [14]) [11,15].

The percentage of resistance toward chloramphenicol recorded in the present study, although moderate, is still alarming since the use of this compound is banned in food-producing animals in all the member states of the European Union that, as expected, reported low levels of resistance in 2017 and 2018 [4]. Although this is speculative, this level of resistance to chloramphenicol could be explained by the illegal and fraudulent use of this antimicrobial in veterinary practices [16].

Interestingly, in our earlier research comparing the results obtained in the years 2003–2007 with those obtained in the subsequent 5 years (2008–2012), we recorded a significantly decreasing trend in the prevalence of resistance to most antimicrobial agents of the survey [17]. What is striking is that the results of the present work showed opposite behavior compared with those obtained in the years 2008–2012, and indeed, an increased resistance to tetracycline, ciprofloxacin, ampicillin, nalidixic acid, cefotaxime, and chloramphenicol among *Salmonella* spp. isolated has been recorded in 2015–2017. Thus, these results show a concerning rapid turnaround of the occurrence of resistance in the same sampling area. Moreover, it is also confirmed by the significant increase in resistance observed for ciprofloxacin within this third period of analysis. The reason for the increased resistance in the latest years may be due to the massive spread of new *S. enterica* clones harboring antimicrobial resistance genes. However, there has been a slight decline in the levels of MDR because, although still high (41.6%), they are lower than those observed in the previous years (82.6% for 2003–2007 and 54.3% for 2008–2012) [17]. 

By serovar, different resistance rates were recorded among the different serotypes of *Salmonella*. Extremely high levels of resistance were observed among *S.* Infantis isolates toward fluoroquinolones and trimethoprim-sulfamethoxazole and, along with *S.* Newport, *S.* Brandenburg, *S.* Rissen, and monophasic *S.* Typhimurium, also against tetracycline and ampicillin. Moreover, one strain among monophasic *S.* Typhimurium isolates was resistant to all the antimicrobials tested. *S.* Newport, *S.* Infantis, and monophasic *S.* Typhimurium belong to the five most commonly reported serovars in human cases, and *S.* Brandenburg has also been frequently reported during confirmed cases of human salmonellosis in the EU [1]. *S.* Rissen, which is frequently reported in the United States of America and Asia, is rarely isolated during human infection in Europe [18]. Since the role of food in *Salmonella* transmission to humans has been demonstrated, the high resistance rate and MDR reported among these serotypes is a matter of great concern [19].

Among *Salmonella* serovars recovered from poultry, overall resistance toward the antimicrobials tested ranged from moderate to very high, except for in response to colistin sulfate and gentamicin. High levels of resistance toward ampicillin, trimethoprim-sulfamethoxazole, tetracycline, and fluoroquinolones were also noted among strains isolated from pork, bovine, and buffalo. These results agree with those reported by the European Food Safety Authority (EFSA) [4]; however, compared to this report, a higher percentage of strains isolated from poultry, pork, and bovine were resistant to at least one antimicrobial class. Moreover, a lower percentage of completely susceptible isolates was reported in the present work. As expected, the occurrence of resistant *Salmonella* strains isolated from molluscan shellfish was significantly lower compared with the occurrence reported for poultry, pork, and bovine. However, the level of resistance reported in the present work was higher than that reported by [20]. Molluscan shellfish are filter-feeding animals, and the presence of resistant *Salmonella* serovars in them may indicate an alteration of water ecosystems by human action [21]. Together with molluscan shellfish, the snails could be also considered sentinels of environmental contamination by resistant bacteria. Snails that may be exposed to resistant bacteria via consumption, for example, of contaminated vegetables, showed a low occurrence of resistant *Salmonella* strains. 

## 4. Materials and Methods

### 4.1. Strains

A total of 178 *Salmonella enterica* strains isolated from 2015 to 2019 in the Campania and Calabria regions of southern Italy were analyzed. Food samples were collected in the context of official controls from public or private enterprises. Except for two poultry samples and one pork samples that were collected frozen and one cooked meat sample of bovine origin, all the other samples did not undergo any preservation process other than chilling. Samples were transported to the lab within one hour and analyzed according to the ISO 6579. In brief, 25-g portions of each sample were homogenized in 225 mL (1:10 (W/W)) buffer peptone water (BPW, CM0509, Oxoid) and incubated at 37 °C for 18 h. Subsequently, 0.1 and 1.0 mL of the incubated homogenates were transferred into Rappaport Vassiliadis broth (RVS, CM0669, Oxoid) and Muller Kaufman broth (MK, CM1048, Oxoid) and incubated at 41.5 °C for 24 h and 37 °C for 24 h, respectively. Then, the enrichments were streaked into xylose-lysinedesoxycholate agar and Salmonella chromogene agar (XLD, CM0469, Oxoid) and incubated at 37 °C for 24 h. Presumptive *Salmonella* colonies were biochemically identified through API 20 E. Afterward, the isolates were serotyped at the *Salmonella* Typing Centre of the Campania Region (Department of Food Microbiology, Istituto Zooprofilattico Sperimentale del Mezzogiorno, Portici, NA, Italy) following the Kaufmann–White scheme (Popoff and Le Minor, 1992) and were assigned into two subspecies and 39 serovars (Table 2). The strains, isolated from foods, were grouped into nine categories, as follows: (1) poultry (meat and meat products), (2) bovine (meat and dairy products), (3) pork (meat and meat products), (4) molluscan shellfish, (5) buffalo (raw milk), (6) ovine (raw milk), (7) snail, (8) vegetables, and (9) mixed meat products (composed of a mix of different ingredients) (Table 3). 

### 4.2. Antibiotic Susceptibility Testing

The antimicrobial susceptibility of the isolates was determined by the disk-diffusion method, following the Clinical and Laboratory Standards Institute (CLSI) recommendations. The following antibiotics (Oxoid, Basingstoke, England, and Becton Dickinson, Mississauga, ON, Canada) were used: nalidixic acid (NAL, 30 μg), ampicillin (AMP, 10 μg), chloramphenicol (CHL, 30 μg), gentamicin (GEN, 10 μg), tetracycline (TET, 30 μg), trimethoprim-sulfamethoxazole (SXT, 25 μg), ciprofloxacin (CIP, 5 μg), colistin sulfate (CST, 10 μg), cephtazidime (CAZ, 10 μg), and cephotaxime (CTX, 30 μg). A quality-control strain (*Escherichia coli* ATCC 25922) was included in the test. The break point for resistance or susceptibility interpretation to each antibiotic was in accordance with the CLSI standards. In the evaluation of the results, the strains displaying intermediate resistance were regarded as resistant, and the strains displaying resistance to at least three antibiotic classes were considered multidrug resistant (MDR) [22,23]. 

### 4.3. Statistical Analysis

The differences in the drug and multidrug resistance between the first two years of the survey (2015–2016) and the second three years (2017–2019), among the different serotypes, and among the different sources were assessed by chi-squared test (χ^2^). The significance of the differences observed was assessed by means of the EpiInfo 7 software package (Centers for Disease Control and Prevention, Atlanta, GA, USA). A value of *p* < 0.05 was accepted as significant. 

## 5. Conclusions

In conclusion, a high prevalence of resistant and MDR *Salmonella* strains among the screened serovars isolated from foods was found. Stringent hygiene and antibiotic stewardship are necessary to prevent an increase in resistant *Salmonella* foodborne infections. Moreover, these data may provide valuable information for developing future *Salmonella* surveillance systems. In particular, high levels of resistance and coresistance toward tetracycline, fluoroquinolones, ampicillin, and trimethoprim-sulfamethoxazole were detected. Furthermore, the results reported in the present work revealed that the antimicrobial resistance trends in the area of southern Italy considered are worryingly changing. Therefore, rapid changes of treatment guidelines and more stringent controls on the use of antimicrobials in veterinary practice are indispensable. 

## Figures and Tables

**Figure 1 antibiotics-09-00365-f001:**
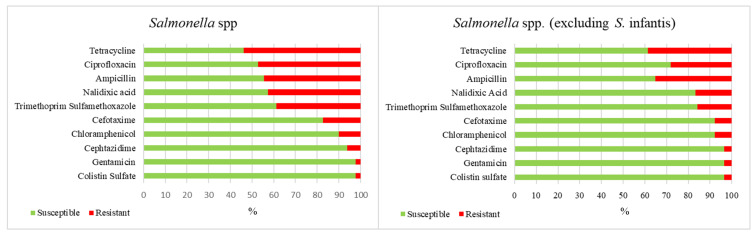
Overall occurrence (%) of susceptibility and resistance to 10 antimicrobials in all *Salmonella* serovars before and after excluding the *S.* Infantis serotype.

**Figure 2 antibiotics-09-00365-f002:**
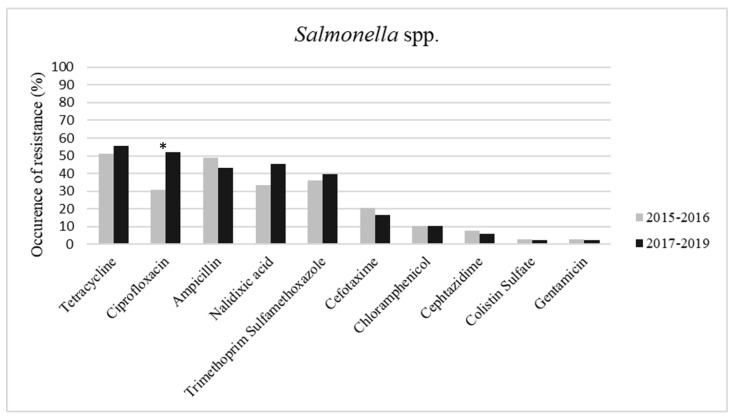
Occurrence (%) of resistance to 10 antimicrobials in *Salmonella* spp. strains isolated in the first two years of survey (2015–2016) and in the second three years (2017–2019). The asterisk (*) indicates the significant differences between the two sampling periods (*p* < 0.05).

**Figure 3 antibiotics-09-00365-f003:**
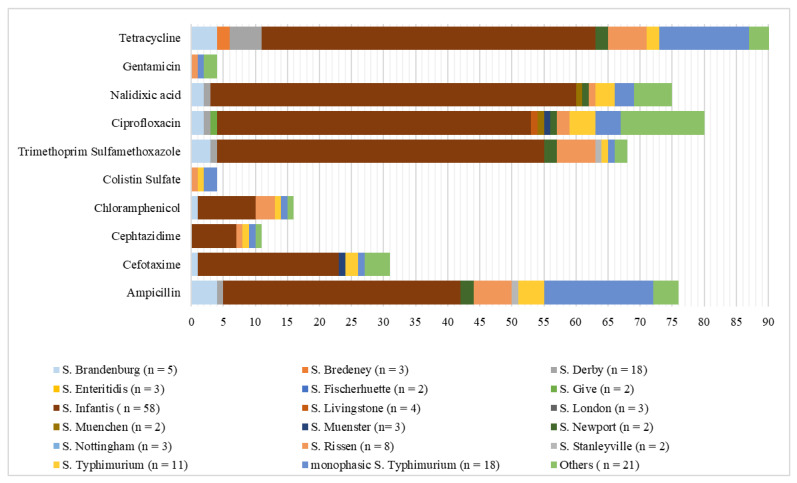
Number of isolates resistant to 10 antimicrobials in *Salmonella* serovars. Serovars with only one strain were summed and are denoted as “Others”. n = total number of isolates per each serovar.

**Figure 4 antibiotics-09-00365-f004:**
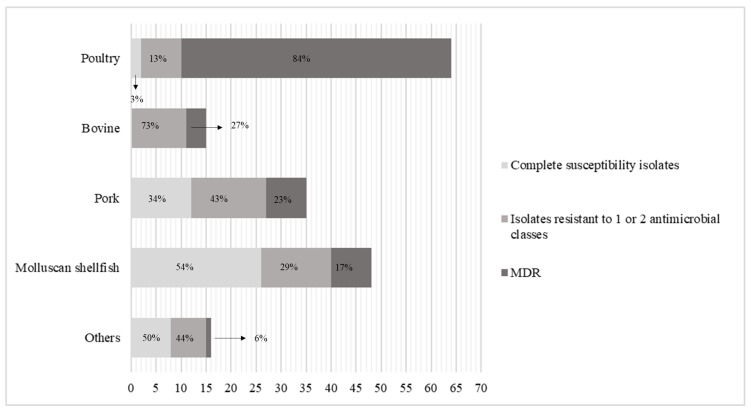
Number and percentage of completely susceptible *Salmonella* isolates, isolates resistant to one or two antimicrobial classes, and multidrug resistant (MDR) isolates from poultry, bovine, pork, and molluscan shellfish. Isolates from buffalo, ovine, snail, vegetables, and mixed meat products were summed and are denoted as “Others”.

**Table 1 antibiotics-09-00365-t001:** Overall occurrence (%) of resistance (R) to 10 antimicrobials in *Salmonella* spp. from poultry (meat and meat products), pork (meat and meat products), molluscan shellfish (MS), bovine (meat and dairy products), buffalo (raw milk), ovine (raw milk), snail, vegetables, and mixed meat products (MMP, composed of a mix of different ingredients). Values in rows bearing different lowercase letters are significantly different (*p* < 0.05).

Antibiotics	Poultry	Pork	MS	Bovine	Buffalo	Ovine	Snail	Vegetable	MMP
Ampicillin	66^a^	49^ac^	21^bd^	53^ae^	20^bce^	0^ad^	33^ad^	0^bce^	0^ad^
Cefotaxime	34^a^	6^b^	10^b^	13^ab^	0^ab^	0^ab^	0^ab^	0^ab^	0^ab^
Cephtazidime	12^a^	6^ac^	2^bc^	0^ac^	0^ac^	0^ac^	0^ac^	0^ac^	0^ac^
Chloramphenicol	14^a^	9^a^	12^a^	0^a^	0^a^	0^a^	0^a^	0^a^	0^a^
Colistin sulfate	0^a^	9^a^	0^a^	7^a^	0^a^	0^a^	0^a^	0^a^	0^a^
Trimethoprim sulfamethoxazole	77^a^	17^bc^	17^bc^	27^bc^	20^bc^	0^ac^	0^ac^	0^bc^	33^ac^
Ciprofloxacin	80^a^	23^bc^	25^bd^	47^ace^	20^bcd^	0^acd^	0^acd^	50^bde^	100^acd^
Nalidixic acid	89^a^	14^b^	10^bc^	33^a^	20^ac^	0^ac^	0^ac^	0^ac^	100^ac^
Gentamicin	0^a^	6^a^	2^a^	0^a^	0^a^	0^a^	0^a^	25^a^	0^a^
Tetracycline	86^ad^	46^bcd^	27^bcd^	73^ac^	0^d^	0^acd^	0^acd^	0^d^	67^acd^

Values in rows bearing different lowercase letters are significantly different (*p* < 0.05).

**Table 2 antibiotics-09-00365-t002:** Number of serovars isolated from 2015 to 2019 used for the evaluation of the antimicrobial resistance. The asterisk (*) indicates not identified serotypes.

Species and Subspecies	Serotype	2015	2016	2017	2018	2019	Total
*S. enterica* subsp. *enterica*	N.I.*				1	1	2
	Infantis	2	7	2	32	16	59
	Derby	5	1	2	9	1	18
	Monophasic *S.* Typhimurium	1		5	10	2	18
	Typhimurium	3	1	1	4	2	11
	Rissen	3	1	1	2	1	8
	Brandenburg		1	2		2	5
	Anatum		1		1	2	4
	Livingstone		1		1	2	4
	Bredeney				2	1	3
	Enteritidis	1		2			3
	London				3		3
	Muenster	1			1	1	3
	Nottingham			1	2		3
	Fischerhuette				2		2
	Give			1		1	2
	Muenchen		1	1			2
	Newport		1			1	2
	Stanleyville		1	1			2
	Agona				1		1
	Blockley				1		1
	Bovismorbificans				1		1
	Carno			1			1
	Eko	1					1
	Goldcoast				1		1
	Havana					1	1
	Hvittingfoss	1					1
	Kapemba	1					1
	Kasenyi				1		1
	Kentucky				1		1
	Litchfield					1	1
	Manchester					1	1
	Mbandaka					1	1
	Mishmarhaemek					1	1
	Panama					1	1
	Pomona					1	1
	Saintpaul	1					1
	Tennessee				1		1
	Winston				1		1
	Worthington	1					1
*S. enterica* subsp. *diarizonae*	N.I.*	2			1		3
	Total	23	16	20	79	40	178

**Table 3 antibiotics-09-00365-t003:** Number of serovars used for the evaluation of the antimicrobial resistance grouped by source. The asterisk (*) indicates not identified serotypes.

Species and Subspecies	Serotype	Poultry	Bovine	Pork	Molluscan Shellfish	Buffalo	Ovine	Snail	Vegetable	Mixed Meat
*S. enterica* subsp. *enterica*	N.I.*		1		1		1			
	Infantis	53	2		3					
	monophasic S. Typhimurium	2	5	8	3					
	Newport	2								1
	Bredeney	1	1		1					
	Derby	1		12	4					
	Livingstone	1			3					
	London	1				2				
	Manchester	1								1
	Saintpaul	1								
	Tennessee	1						1		
	Agona		1							
	Anatum				4					
	Blockley									
	Bovismorbificans					1				
	Brandenburg			1	4				1	
	Carno		1							1
	Eko		1							
	Enteritidis				2				1	
	Fischerhuette				2					
	Give			1	1					
	Goldcoast			1						
	Havana				1					
	Hvittingfoss									
	Kapemba		1							
	Kasenyi									
Species and Subspecies	Serotype	Poultry	Bovine	Pork	Molluscan shellfish	Buffalo	Ovine	Snail	Vegetable	Mixed Meat
*S. enterica* subsp. *enterica*	Kentucky				1					
	Litchfield				1					
	Mbandaka				1					
	Mishmarhaemek			1						
	Muenchen			1		1			1	
	Muenster		1		2					
	Nottingham				2					
	Panama				1					
	Pomona				1					
	Rissen		1	4	3					
	Stanleyville				1	1				
	Typhimurium			6	5					
	Winston								1	
	Worthington				1					
*S. enterica* subsp. *diarizonae*	N.I.*							2		
	Total	64	15	35	48	5	1	3	4	3

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
