# Peer review of "Antimicrobial Susceptibility Testing for Salmonella Serovars Isolated from Food Samples: Five-Year Monitoring (2015–2019)"

_antibiotics, 2020, doi:10.3390/antibiotics9070365_

Round 1
Reviewer 1 Report
In the present study, Peruzy et al, demonstrated the incidence of antimicrobial resistance among Salmonella serovars isolated from different sources of food in the period of 2015-2019. The authors have also tested the sensitivity and resistance of total 178 Salmonella strains belonging to 25 serovars against ten antimicrobials. The authors have observed that high proportions of Salmonella isolates were resistant to tetracycline (n = 53.9%), ciprofloxacin (n = 47.2%), ampicillin (n = 44.4%), nalidixic acid 17 (n = 42.7%) and trimethoprim-sulfamethoxazole (n = 38.8%). Additionally, the authors have represented the overall occurrence (%) of susceptibility (S) and resistance (R) to 10 antimicrobials in Salmonella spp. in different food sources. The manuscript is well written and presented their observation considerately. However, there are some concerns that need to be addressed.
Comments:
- The statistical analysis need to be more specific.
- Need to provide the food sources more specifically (i.e. frozen or freshly collected for vegetables/meat) and also need to mention the way they have collected.
- Authors did not mention in the methodology how they isolated the bacteria? What are the growing conditions?
- The authors have compared between the sampling periods 2015–2016 and 2017–2019 and showed an increase in resistance to four single drugs (tetracycline, ciprofloxacin, nalidixic acid, and trimethoprim sulfamethoxazole). How will the authors interpret this? What could be the reasons for the increase in resistance?
Author Response
Dear Review,
We thank the reviewers for their valuable comments.
We made the necessary adaptations and corrections throughout the manuscript (highlighted in yellow).
Point-by-point response:
In the present study, Peruzy et al, demonstrated the incidence of antimicrobial resistance among Salmonella serovars isolated from different sources of food in the period of 2015-2019. The authors have also tested the sensitivity and resistance of total 178 Salmonella strains belonging to 25 serovars against ten antimicrobials. The authors have observed that high proportions of Salmonella isolates were resistant to tetracycline (n = 53.9%), ciprofloxacin (n = 47.2%), ampicillin (n = 44.4%), nalidixic acid 17 (n = 42.7%) and trimethoprim-sulfamethoxazole (n = 38.8%). Additionally, the authors have represented the overall occurrence (%) of susceptibility (S) and resistance (R) to 10 antimicrobials in Salmonella spp. in different food sources. The manuscript is well written and presented their observation considerately. However, there are some concerns that need to be addressed.
POINT 1: The statistical analysis need to be more specific.
Response 1: As suggested by the reviewer we have carried out a more specific statistical analysis. Please refer to table 1 and to lines 119-123 and 259-261. Moreover, thanks to thorough review, we stumbled upon a minor calculation mistake in our data (highlight in green) at lines 92 and 185.
POINT 2: Need to provide the food sources more specifically (i.e. frozen or freshly collected for vegetables/meat) and also need to mention the way they have collected.
Response 2: To clarify we added the following part “Food samples were collected in the context of official controls from public or private enterprises. Except for two poultry and one pork samples that were collected frozen and one cooked meat sample of bovine origin, all the other did not undergo any preserving process other than chilling” at lines 219-222.
POINT 3: Authors did not mention in the methodology how they isolated the bacteria? What are the growing conditions?
Response 3: We agree with the reviewer, and therefore the isolation method has now been included in Material and Methods. Please refer to lines 222-230.
POINT 4: The authors have compared between the sampling periods 2015–2016 and 2017–2019 and showed an increase in resistance to four single drugs (tetracycline, ciprofloxacin, nalidixic acid, and trimethoprim sulfamethoxazole). How will the authors interpret this? What could be the reasons for the increase in resistance?
Response 4: Although it is speculative, in our opinion the reason for the increased resistance in the latest years may be due to the massive spread of new S. enterica clones harboring antimicrobial resistance genes. This statement has now been included at lines 185-187.
Yours sincerely,
Yolande Therese Rose Proroga, corresponding author

Reviewer 2 Report
The paper reports antimicrobial susceptibility of Salmonella serovars isolated from food samples between 2015 and 2019. This is a significant study and the manuscript is well written. I think this paper can be accepted for publication after revising the following minor points.
L13. problematic -> problem
Table 1. I do not think that both of susceptibility (S) and resistance (R) need to be shown. It is enough to show one of them. Also, values after the decimal point should be removed.
Fig 4. Values after the decimal point should be removed.
L148. fluoroquinolones, are -> fluoroquinolones are
L149. standard treatment for invasive -> standard for treatment against invasive (?)
L151-153. Please rewrite the sentence.
L192. one, except -> one, and except (?)
L194-195. were also noted among isolated recovered -> was also noted among strains isolated
L203. also the snails could be considered -> the snails could be also considered
L213 & Table 2. Since 39 serovars are listed in Table 2, ‘25 serovars’ in line 213 should be replaced with ‘39 serovars’.
Table 2. Enterica -> enterica, Diarizonae -> diarizonae
L231. ‘Escherichia coli’ should be shown in italic format.
L246. was -> were
L246-249. Please rewrite the sentence.
Author Response
Dear Reviewer,
We thank the reviewers for their valuable comments.
We made the necessary adaptations and corrections throughout the manuscript (highlighted in yellow).
Point-by-point response:
The paper reports antimicrobial susceptibility of Salmonella serovars isolated from food samples between 2015 and 2019. This is a significant study and the manuscript is well written. I think this paper can be accepted for publication after revising the following minor points.
POINT 1: L13. problematic -> problem
Response 1: The word “problematic” has been replaced with “problem”. Please refer to line 13
POINT 2: Table 1. I do not think that both of susceptibility (S) and resistance (R) need to be shown. It is enough to show one of them. Also, values after the decimal point should be removed.
Response 2: We agree with the reviewer and we have removed the results concerning the susceptibility from Table 1 and the values after the decimal point.
POINT 3: Fig 4. Values after the decimal point should be removed.
Response 3: We agree with the reviewer and we have removed the values after the decimal point in Figure 4
POINT 4: L148. fluoroquinolones, are -> fluoroquinolones are
Response 4: The comma after “fluoroquinolones” has been removed. Please refer to line 155.
POINT 5: L149. standard treatment for invasive -> standard for treatment against invasive (?)
Response 5: As suggested by the reviewer “standard treatment for invasive” has been replaced with “standard for treatment against invasive”. Please refer to line 156.
POINT 6: L151-153. Please rewrite the sentence.
Response 6: The sentence at line 158-160 has been rewritten.
POINT 7: L192. one, except -> one, and except (?)
Response 7: The sentence has been rewritten. Please refer to lines 201-202.
POINT 8: L194-195. were also noted among isolated recovered -> was also noted among strains isolated
Response 8: As suggested by the reviewer “were also noted among isolated recovered” has been replaced with “was also noted among strains isolated”. Please refer to line 204.
POINT 9: L203. also the snails could be considered -> the snails could be also considered
Response 9: As suggested by the reviewer “also the snails could be considered” has been replaced with “the snails could be also considered”. Please refer to lines 212-213.
POINT 10: L213 & Table 2. Since 39 serovars are listed in Table 2, ‘25 serovars’ in line 213 should be replaced with ‘39 serovars’.
Response 10: We agree with the reviewer, this was a mistake. The correct number has now been included in the text. Please refer to line 233.
POINT 11: Table 2. Enterica -> enterica, Diarizonae -> diarizonae
Response 11: in Table 2 and Table 3 Enterica and Diarizonae are now in italic
POINT 12: L231. ‘Escherichia coli’ should be shown in italic format.
Response 12: “Escherichia coli” is now in italic. Please refer to line 251.
POINT 13: L246. was -> were
Response 13: “Was” has been replaced with “were”. Please refer to line 268.
POINT 14: L246-249. Please rewrite the sentence
Response 14: The sentences (268-271) are now rewritten as “Furthermore, the results reported in the present work revealed that the antimicrobial resistance trends in the arear of southern Italy considered are worryingly changing. Therefore, rapid changes of the treatments guidelines and more stringent controls on the use of antimicrobials in the veterinary practice are indispensable”
Yours sincerely,
Yolande Therese Rose Proroga, corresponding author

Reviewer 3 Report
Dear Authors
Thank you for your very important research paper. Overall, the manuscript looks very good and sound. There are just a few minor suggestions for you to consider for clarity.
Line 13. “Problematic” should be changed to “problem”.
Line 20. “At” should be changed to “resistance to at least”……..
Line 24. Consider removing “showed to be” and add “study indicates a potential public health risk”. “Consequently, additional hygiene and antibiotic stewardship practices should be considered for the food industry to prevent the prevalence of Salmonella in foods.”
Line 30. Replace “cause by” with “due to”
Line 38. Remove “respectively”
Fig 2. Can you change the colors so they are more distinct when printed in Black and White? Can you place notations on the columns that represent significant differences?
Line 142. Salmonella spp. Indicates many different species. However, Salmonella only really has one species and many serotypes. Consider changing to “Salmonella serotypes”.
Lines 191-193. This sentence is long and awkward. Please break apart into two sentences to enhance the clarity, thank you.
Line 195-198. Sentence starting with “These results”…………
Also very long and unclear. Consider breaking into two sentences to enhance the clarity, thank you.
Line 203. Remove “also the”… “shellfish, snails could also be considered……….”.
Lines 242-243. Starting at “To avoid………….” Add “stringent hygiene and antibiotic stewardship in necessary to prevent”……………
Thank you .
Author Response
Dear Reviewer,
We thank the reviewers for their valuable comments.
We made the necessary adaptations and corrections throughout the manuscript (highlighted in yellow).
Dear Authors, Thank you for your very important research paper. Overall, the manuscript looks very good and sound. There are just a few minor suggestions for you to consider for clarity.
POINT 1: Line 13. “Problematic” should be changed to “problem”.
Response 1: The word “problematic” has been replaced with “problem”. Please refer to line 13
POINT 2: Line 20. “At” should be changed to “resistance to at least”……..
Response 2: “At” has been replaced with “resistance to at least”. Please refer to line 20
POINT 3: Line 24. Consider removing “showed to be” and add “study indicates a potential public health risk”. “Consequently, additional hygiene and antibiotic stewardship practices should be considered for the food industry to prevent the prevalence of Salmonella in foods.”
Response 3: As suggested by the reviewer sentences at lines 24-26 have been modified as “The high levels of resistance reported in the present study indicate a potential public health risk. Consequently, additional hygiene and antibiotic stewardship practices should be considered for the food industry to prevent the prevalence of Salmonella in foods.”
POINT 4: Line 30. Replace “cause by” with “due to”
Response 4: “cause by” has been replaced with “due to”. Please refer to line 30
POINT 5: Line 38. Remove “respectively”
Response 5: The word “respectively “at line 38 has been removed.
POINT 6: Fig 2. Can you change the colors so they are more distinct when printed in Black and White? Can you place notations on the columns that represent significant differences?
Response 6 : Thanks to thorough review, we stumbled upon a minor calculation mistake in our data (highlight in green) at lines 92 and 185. Moreover, as suggested by the reviewer the colors of figure 2 have been changed and an asterisk have been added to indicate significant differences.
POINT 7: Line 142. Salmonella spp. Indicates many different species. However, Salmonella only really has one species and many serotypes. Consider changing to “Salmonella serotypes”.
Response 7: We agree with the reviewer and we have replaced “spp.” with “serotypes”. Please refer to line 149.
POINT 8: Lines 191-193. This sentence is long and awkward. Please break apart into two sentences to enhance the clarity, thank you.
Response 8: As suggested by the reviewer the sentence (201-202) is now rewritten as “Among Salmonella serovars recovered from poultry overall resistance towards the antimicrobial tested ranged from moderate to very high, except for colistin sulfate and gentamicin”
POINT 9: Line 195-198. Sentence starting with “These results”…………
Also very long and unclear. Consider breaking into two sentences to enhance the clarity, thank you.
Response 9: As suggested by the reviewer the sentences (204-207) are now rewritten as ”These results agree with those reported by EFSA [4], however, compared to this report a higher percentage of strains isolated from poultry, pork, and bovine was resistant to at least one antimicrobial classes. Moreover, a lower percentage of complete susceptible isolates were reported in the present work”.
POINT 10: Line 203. Remove “also the”… “shellfish, snails could also be considered……….”.
Response 10: The sentence at lines 212-213 has been rewritten as “Together with molluscan shellfish, the snails could be also considered as sentinel of the environmental contamination by resistant bacteria.”
POINT 11: Lines 242-243. Starting at “To avoid………….” Add “stringent hygiene and antibiotic stewardship in necessary to prevent”……………
Response 11: As suggested by the reviewer the sentence (264-265) is now rewritten as “Stringent hygiene and antibiotic stewardship are necessary to prevent an increase in resistant Salmonella foodborne infections”
Yours sincerely,
Yolande Therese Rose Proroga, corresponding author

Reviewer 4 Report
The paper by Peruzy (authors=Au) investigates the occurrence of antimicrobial resistance among Salmonella serovars collected from 2015 to 2019 from different food commodities, in the Campania and Calabria regions of southern Italy. Salmonella is a major foodborne pathogen and an improved understanding of the distribution of antibiotic resistance patterns is a research that is of merit. While this study addresses an important topic to the "Antibiotics", unfortunately, there are serious concerns with the study design.
Major Comments
First of all, the selection of isolates is completely random. The AUs compare Salmonella strains isolated from foods belonging into nine categories: (i) Poultry (meat and meat products), (ii) Bovine (meat and dairy products), (iii) Pork (meat and meat products), (iv) Molluscan shellfish, (v) Buffalo (raw milk), (vi) Ovine (raw milk), (vii) Snail, (viii) Vegetable and (ix) mixed meat products (composed by a mix of different ingredients). Furthermore, the set does not contain any clinical isolates from humans, it would have been interesting to compare antibiotic resistance profiles (ARPs).
Secondly, the study does not contain any molecular characterization for Salmonella isolates (screening for virulence and resistance genes). It would have been interesting to investigate the distribution of virulence genes among Salmonella serovars and whether there is any relationship between ARPs and virulence determinants
Third, some analysis misses a rationale. Why to study the occurrence of susceptibility after excluding the S. Infantis serotype (Figure 1)?
Unfortunately, I must conclude that the missing novelty and poor design of the study does not warrant publication in the present form in an internationally acknowledged journal with a broad readership. However, to foster the exchange of knowledge among public health authorities, to whom the data might be of some interest, and the AU in Italy, I recommend to publish some parts at a national level.
Minor comments
1. There are some issues with the life science data writing. A few examples where data is lacking clarity or reads strangely:
a. Lines 17-18: (n=53,9%); (n=47,2%)....
b. Line 43: 700,000 death instead of 700,00
2. References (numbered 6, 12, 17, 22). For references, authors should follow the instructions given in the Manuscript Preparation of the journal
3. Lines 83-89. ARPs of salmonella isolates are discussed. However, it would be convenient for their distribution to be presented in a table.
4. Line 103. Figure 3 does not give information for the ratio of resistance to antibiotics but the number of isolates resistant to antibiotics
5. Table 2 is a redundancy
6. Lines 236-239. In the statistical analysis there is no description for the determination of the significant differences in the occurrence of resistance to the antimicrobials tested (referred in lines 128-137)
Author Response
Dear Reviewer,
We thank the reviewers for their valuable comments.
We made the necessary adaptations and corrections throughout the manuscript (highlighted in yellow).
Reviewer 1
Reviewer: In the present study, Peruzy et al, demonstrated the incidence of antimicrobial resistance among Salmonella serovars isolated from different sources of food in the period of 2015-2019. The authors have also tested the sensitivity and resistance of total 178 Salmonella strains belonging to 25 serovars against ten antimicrobials. The authors have observed that high proportions of Salmonella isolates were resistant to tetracycline (n = 53.9%), ciprofloxacin (n = 47.2%), ampicillin (n = 44.4%), nalidixic acid 17 (n = 42.7%) and trimethoprim-sulfamethoxazole (n = 38.8%). Additionally, the authors have represented the overall occurrence (%) of susceptibility (S) and resistance (R) to 10 antimicrobials in Salmonella spp. in different food sources. The manuscript is well written and presented their observation considerately. However, there are some concerns that need to be addressed.
POINT 1: The statistical analysis need to be more specific.
Response 1: As suggested by the reviewer we have carried out a more specific statistical analysis. Please refer to table 1 and to lines 119-123 and 259-261. Moreover, thanks to thorough review, we stumbled upon a minor calculation mistake in our data (highlight in green) at lines 92 and 185.
POINT 2: Need to provide the food sources more specifically (i.e. frozen or freshly collected for vegetables/meat) and also need to mention the way they have collected.
Response 2: To clarify we added the following part “Food samples were collected in the context of official controls from public or private enterprises. Except for two poultry and one pork samples that were collected frozen and one cooked meat sample of bovine origin, all the other did not undergo any preserving process other than chilling” at lines 219-222.
POINT 3: Authors did not mention in the methodology how they isolated the bacteria? What are the growing conditions?
Response 3: We agree with the reviewer, and therefore the isolation method has now been included in Material and Methods. Please refer to lines 222-230.
POINT 4: The authors have compared between the sampling periods 2015–2016 and 2017–2019 and showed an increase in resistance to four single drugs (tetracycline, ciprofloxacin, nalidixic acid, and trimethoprim sulfamethoxazole). How will the authors interpret this? What could be the reasons for the increase in resistance?
Response 4: Although it is speculative, in our opinion the reason for the increased resistance in the latest years may be due to the massive spread of new S. enterica clones harboring antimicrobial resistance genes. This statement has now been included at lines 185-187.
Reviewer 2
The paper reports antimicrobial susceptibility of Salmonella serovars isolated from food samples between 2015 and 2019. This is a significant study and the manuscript is well written. I think this paper can be accepted for publication after revising the following minor points.
POINT 1: L13. problematic -> problem
Response 1: The word “problematic” has been replaced with “problem”. Please refer to line 13
POINT 2: Table 1. I do not think that both of susceptibility (S) and resistance (R) need to be shown. It is enough to show one of them. Also, values after the decimal point should be removed.
Response 2: We agree with the reviewer and we have removed the results concerning the susceptibility from Table 1 and the values after the decimal point.
POINT 3: Fig 4. Values after the decimal point should be removed.
Response 3: We agree with the reviewer and we have removed the values after the decimal point in Figure 4
POINT 4: L148. fluoroquinolones, are -> fluoroquinolones are
Response 4: The comma after “fluoroquinolones” has been removed. Please refer to line 155.
POINT 5: L149. standard treatment for invasive -> standard for treatment against invasive (?)
Response 5: As suggested by the reviewer “standard treatment for invasive” has been replaced with “standard for treatment against invasive”. Please refer to line 156.
POINT 6: L151-153. Please rewrite the sentence.
Response 6: The sentence at line 158-160 has been rewritten.
POINT 7: L192. one, except -> one, and except (?)
Response 7: The sentence has been rewritten. Please refer to lines 201-202.
POINT 8: L194-195. were also noted among isolated recovered -> was also noted among strains isolated
Response 8: As suggested by the reviewer “were also noted among isolated recovered” has been replaced with “was also noted among strains isolated”. Please refer to line 204.
POINT 9: L203. also the snails could be considered -> the snails could be also considered
Response 9: As suggested by the reviewer “also the snails could be considered” has been replaced with “the snails could be also considered”. Please refer to lines 212-213.
POINT 10: L213 & Table 2. Since 39 serovars are listed in Table 2, ‘25 serovars’ in line 213 should be replaced with ‘39 serovars’.
Response 10: We agree with the reviewer, this was a mistake. The correct number has now been included in the text. Please refer to line 233.
POINT 11: Table 2. Enterica -> enterica, Diarizonae -> diarizonae
Response 11: in Table 2 and Table 3 Enterica and Diarizonae are now in italic
POINT 12: L231. ‘Escherichia coli’ should be shown in italic format.
Response 12: “Escherichia coli” is now in italic. Please refer to line 251.
POINT 13: L246. was -> were
Response 13: “Was” has been replaced with “were”. Please refer to line 268.
POINT 14: L246-249. Please rewrite the sentence
Response 14: The sentences (268-271) are now rewritten as “Furthermore, the results reported in the present work revealed that the antimicrobial resistance trends in the arear of southern Italy considered are worryingly changing. Therefore, rapid changes of the treatments guidelines and more stringent controls on the use of antimicrobials in the veterinary practice are indispensable”
Reviewer 3
Dear Authors, Thank you for your very important research paper. Overall, the manuscript looks very good and sound. There are just a few minor suggestions for you to consider for clarity.
POINT 1: Line 13. “Problematic” should be changed to “problem”.
Response 1: The word “problematic” has been replaced with “problem”. Please refer to line 13
POINT 2: Line 20. “At” should be changed to “resistance to at least”……..
Response 2: “At” has been replaced with “resistance to at least”. Please refer to line 20
POINT 3: Line 24. Consider removing “showed to be” and add “study indicates a potential public health risk”. “Consequently, additional hygiene and antibiotic stewardship practices should be considered for the food industry to prevent the prevalence of Salmonella in foods.”
Response 3: As suggested by the reviewer sentences at lines 24-26 have been modified as “The high levels of resistance reported in the present study indicate a potential public health risk. Consequently, additional hygiene and antibiotic stewardship practices should be considered for the food industry to prevent the prevalence of Salmonella in foods.”
POINT 4: Line 30. Replace “cause by” with “due to”
Response 4: “cause by” has been replaced with “due to”. Please refer to line 30
POINT 5: Line 38. Remove “respectively”
Response 5: The word “respectively “at line 38 has been removed.
POINT 6: Fig 2. Can you change the colors so they are more distinct when printed in Black and White? Can you place notations on the columns that represent significant differences?
Response 6 : Thanks to thorough review, we stumbled upon a minor calculation mistake in our data (highlight in green) at lines 92 and 185. Moreover, as suggested by the reviewer the colors of figure 2 have been changed and an asterisk have been added to indicate significant differences.
POINT 7: Line 142. Salmonella spp. Indicates many different species. However, Salmonella only really has one species and many serotypes. Consider changing to “Salmonella serotypes”.
Response 7: We agree with the reviewer and we have replaced “spp.” with “serotypes”. Please refer to line 149.
POINT 8: Lines 191-193. This sentence is long and awkward. Please break apart into two sentences to enhance the clarity, thank you.
Response 8: As suggested by the reviewer the sentence (201-202) is now rewritten as “Among Salmonella serovars recovered from poultry overall resistance towards the antimicrobial tested ranged from moderate to very high, except for colistin sulfate and gentamicin”
POINT 9: Line 195-198. Sentence starting with “These results”…………
Also very long and unclear. Consider breaking into two sentences to enhance the clarity, thank you.
Response 9: As suggested by the reviewer the sentences (204-207) are now rewritten as ”These results agree with those reported by EFSA [4], however, compared to this report a higher percentage of strains isolated from poultry, pork, and bovine was resistant to at least one antimicrobial classes. Moreover, a lower percentage of complete susceptible isolates were reported in the present work”.
POINT 10: Line 203. Remove “also the”… “shellfish, snails could also be considered……….”.
Response 10: The sentence at lines 212-213 has been rewritten as “Together with molluscan shellfish, the snails could be also considered as sentinel of the environmental contamination by resistant bacteria.”
POINT 11: Lines 242-243. Starting at “To avoid………….” Add “stringent hygiene and antibiotic stewardship in necessary to prevent”……………
Response 11: As suggested by the reviewer the sentence (264-265) is now rewritten as “Stringent hygiene and antibiotic stewardship are necessary to prevent an increase in resistant Salmonella foodborne infections”
Yours sincerely,
Yolande Therese Rose Proroga, corresponding author

Round 2
Reviewer 4 Report
The paper by Peruzy (authors=Au) investigates the occurrence of antimicrobial resistance among Salmonella serovars collected from 2015 to 2019 from different food commodities, in the Campania and Calabria regions of southern Italy. Salmonella is a major foodborne pathogen and an improved understanding of the distribution of antibiotic resistance patterns is a research that is of merit. While this study addresses an important topic to the "Antibiotics", unfortunately, there are serious concerns with the study design.
Major Comments
First of all, the selection of isolates is completely random. The AUs compare Salmonella strains isolated from foods belonging into nine categories: (i) Poultry (meat and meat products), (ii) Bovine (meat and dairy products), (iii) Pork (meat and meat products), (iv) Molluscan shellfish, (v) Buffalo (raw milk), (vi) Ovine (raw milk), (vii) Snail, (viii) Vegetable and (ix) mixed meat products (composed by a mix of different ingredients). Furthermore, the set does not contain any clinical isolates from humans, it would have been interesting to compare antibiotic resistance profiles (ARPs).
Secondly, the study does not contain any molecular characterization for Salmonella isolates (screening for virulence and resistance genes). It would have been interesting to investigate the distribution of virulence genes among Salmonella serovars and whether there is any relationship between ARPs and virulence determinants
Third, some analysis misses a rationale. Why to study the occurrence of susceptibility after excluding the S. Infantis serotype (Figure 1)?
Unfortunately, I must conclude that the missing novelty and poor design of the study does not warrant publication in the present form in an internationally acknowledged journal with a broad readership. However, to foster the exchange of knowledge among public health authorities, to whom the data might be of some interest, and the AU in Italy, I recommend to publish some parts at a national level.
Minor comments
1. There are some issues with the life science data writing. A few examples where data is lacking clarity or reads strangely:
a. Lines 17-18: (n=53,9%); (n=47,2%)....
b. Line 43: 700,000 death instead of 700,00
2. References (numbered 6, 12, 17, 22). For references, authors should follow the instructions given in the Manuscript Preparation of the journal
3. Lines 83-89. ARPs of salmonella isolates are discussed. However, it would be convenient for their distribution to be presented in a table.
4. Line 103. Figure 3 does not give information for the ratio of resistance to antibiotics but the number of isolates resistant to antibiotics
5. Table 2 is a redundancy